# Training Methods Used by Dog Guardians in the United States: Prevalence, Sources of Information, and Reasons for Use

**DOI:** 10.3390/ani14091310

**Published:** 2024-04-27

**Authors:** Anamarie C. Johnson, Clive D. L. Wynne

**Affiliations:** Department of Psychology, Arizona State University, P.O. Box 871104, Tempe, AZ 85287, USA; cwynne1@asu.edu

**Keywords:** dog training, human-dog interaction, behavior concerns

## Abstract

**Simple Summary:**

Recent research has focused on the welfare implications of differing dog training methodologies, but there is less insight into how dog guardians may train their dogs. The aim of this survey study was to gain insight into what training methods dog guardians use, particularly the use of aversive methods, and where they received their recommendations.

**Abstract:**

While there has been recent attention in the scientific community on the ethical and welfare implications of different dog training methods, less research has investigated what methods and training tools United States dog guardians use, where they obtain information about dog training, and the reasons they give for their choices of method. We conducted two surveys with nearly 800 Arizona State University undergraduate students to gain a more realistic look into how dog guardians in the United States train their dogs and where they are receiving their training information. Only 5% of respondents reported utilizing a trainer when they had concerns regarding their dog’s behavior; 60% would ask a friend or family member or seek advice online. Few reported taking their dog to any training classes; 70% reported either training the dog themselves or not implementing any formal training. When asked general questions, most respondents reported using rewards-based methods but, when asked about specific problem behaviors, 57% of respondents noted that they would use auditory or physical corrections. Respondents who trained with rewards-based methods reported that these methods of training were more effective significantly more frequently than those who trained with mixed methods reported that those methods were most effective (Fisher’s Exact Test, *p* < 0.01).

## 1. Introduction

In 2023, an estimated 65 million households in the United States owned a dog and spent over $136 billion on their pets [1]. Since the COVID19 epidemic started in 2020, the US dog training industry has seen a rise of $8 million in revenue and is projected to earn over $820 million by 2026 [2]. As dog ownership increases, more families must navigate how best to train their dogs and deal with behavior challenges.

Dog owners, particularly in the United States, have a wide range of training methodologies available to them. Most dog trainers can be divided between two camps: those that utilize only non-aversive methods and those that utilize a mix of non-aversive and aversive methods [3]. Each of these camps is rooted in the work of B.F. Skinner who divided operant conditioning into four quadrants. Non-aversive trainers use two of the four quadrants: positive reinforcement (adding a rewarding stimulus to increase the likelihood of a behavior occurring again) and negative punishment (removing a rewarding stimulus to decrease the likelihood of a behavior occurring again). Aversive trainers use these two quadrants but also the remaining two: positive punishment (adding an unpleasant stimulus to decrease the likelihood a behavior will occur again) and negative reinforcement (removing an unpleasant stimulus to increase the likelihood a behavior will occur again [3].

Positive punishment with electronic shock collars (e-collars) has been shown effective in stopping problematic chasing behavior [4,5], but the use of e-collars did not lead to significantly higher rates of owner-reported success than solely rewards-based (non-aversive) methods in a study of sheep chasing by Cooper and colleagues [6]. Dogs trained with aversive methods have often shown more stress-related behaviors such as yawning, backward-oriented ears, or avoidance behaviors than those in non-aversive conditions [7,8,9,10].

Beyond immediate welfare effects, dogs exposed to aversive training methods were less successful at completing a novel training task and showed a more pessimistic bias on a spatial cognitive bias test than those trained with non-aversive methods [10,11]. Dogs trained with aversive methods were less interactive during play solicited by an owner and were less likely to interact with a stranger in a relaxed environment [11]. Similarly, dogs trained with aversive methods spent less time gazing at their owners during training and showed avoidance behavior whereas those in non-aversive conditions did not [8].

Partially because of these findings, many countries have implemented or at least considered bans on aversive training tools, particularly e-collars. Eleven European countries have banned the use of e-collars and France is in the process of implementing a legal ban [12,13]. In the Australian state of Victoria, e-collars can only be used by experienced animal professionals like veterinarians or skilled dog trainers [12]. In New Zealand, where dogs chasing and killing endangered wildlife is a concern [5], e-collars can only be utilized if other methods have been unsuccessful, and the dog is at risk of euthanasia. Prong collars are banned in four European countries and in New Zealand and Victoria [12].

There are no legal bans or legislation restricting the use of aversive tools in the United States; additionally, there is no formal licensure requirement for practicing dog trainers. However, many U.S. organizations strongly advocate for the use of non-aversive training [14,15], and in 2020, the major pet product retailer, Petco, stopped selling e-collars in their stores [16].

Despite these political and social moves to restrict aversive training, previous research has shown that some dog owners still rely on aversive methods to tackle behavior problems. Blackwell and colleagues [17] found that 3% of dog owners in the United Kingdom reported using a remote e-collar, 1.4% used a bark-activated collar and 0.9% used an invisible-fence-activated collar. However, the authors noted that responses might have been influenced by the political movements to ban such devices in the U.K. In France, where e-collars were not yet banned during 2015 data collection, 26% of respondents reported using such devices, and nearly 72% used them without any professional advice [18]. In the United States, in a study of over 1000 adolescent dog owners, 2% reported using an e-collar with their dogs [19].

Using a convenience sample of dog owners in the United Kingdom walking their dogs or visiting a veterinarian, Blackwell and colleagues found that only 16% of owners reported consistently using positive reinforcement (non-aversive) training and 72% of all respondents used some form of positive punishment (aversive) in their daily lives [20]. Another study of UK owners, also sampling owners walking their dogs, showed a similar trend with over 75% of respondents saying they utilized rewards-based (non-aversive) training but that often included a combination of positive reinforcement with aversive methods [21].

Some prior surveys have investigated where owners acquire their training information and recommendations. One study found that while nearly all survey respondents reported their dog having some sort of undesirable behavior, only 18% sought help [20]. Of those, 47% sought the advice of a dog trainer and 9% asked a friend or relative, often someone who was experiencing similar problems with their own dog [20]. Eighty-eight percent of respondents in this survey reported doing some training with their dog, but 58% of owners trained the dogs themselves [20]. This matches the findings of Herron and colleagues who noted that the most common resource for a problem behavior was the owner relying on his or her past experience followed by seeking the advice of a dog trainer [22]. As the authors noted, given that most owners are not skilled in behavior modification, this presents a safety risk to the owner and the dog [22].

The aim of the current study was to gather information about how American families choose to train their dogs, what training tools they use, as well as how they handle common problem behaviors, and from whom they seek training advice. Of particular interest were the aversive tools and methods owners use to tackle behavior problems, whether recommendations for training approaches were coming from professional sources and, overall, whether owners viewed aversive techniques as effective.

## 2. Materials and Methods

### 2.1. Subjects

The survey was provided as an optional extra credit opportunity to students taking any introductory psychology course at Arizona State University in the Spring or Fall semesters of 2021. All responses were anonymized through the online survey instrument Qualtrics (Qualtrics, Provo, UT, USA). In spring, a total of 410 participants engaged with the survey between 19 January 2021, and 15 March 2021. Of these, 12 participants were removed due to incomplete information, leaving 398 respondents. In fall, a total of 438 submissions occurred between 6 September 2021, and 23 September 2021. Of those responses, 40 were removed due to incomplete information, leaving 398 complete submissions.

### 2.2. Survey

The survey administered during Spring 2021 consisted of up to 42 questions and the Fall 2021 survey consisted of up to 44 questions (Appendix A). The number of questions presented to each participant varied because certain responses led to follow-up questions.

After giving voluntary consent to participate, the first portion of the survey contained four questions related to demographics. The second portion contained four questions about where the dog resided, the number of individuals in the home, number of other dogs in the home, and the survey respondent’s relationship to the dog. The third section consisted of five questions relating to the dog. Respondents reported on the dog they spent the most time with or, if they could not distinguish between dogs on that dimension, to select the dog whose name came first in the alphabet. Questions were asked of the dog’s birthdate, age, sex, and where the respondent first heard about the dog and, ultimately, where they acquired the dog.

In Spring 2021, the fourth portion of the survey consisted of a possible ten questions related to dog leash walking equipment, training class experience, and the training methodology of those classes. Respondents were asked where they might obtain dog behavior information. In Fall 2021, a question was added asking whether students obtained information from social media sources like Facebook, Instagram, Twitter, or TikTok.

The final portion of the survey consisted of a possible 19 questions. Eight questions related to common behavior problems: barking in the home, destructive behavior, aggression to dogs, jumping on people, leash pulling, not coming when called, aggression to people, and resource guarding. For each of these behavior problems, several possible training techniques were suggested (Appendix A). If a respondent selected a specific technique, they were directed to a question about who recommended that technique.

After these focused behavior questions, respondents were asked which training methodology they thought was least or most effective. For the technique that they identified as most effective, they were asked to select no more than two options for why they thought that technique worked; options included ease of use, tolerance by the dog, inexpensiveness, and that the dog does not show the behavior anymore.

### 2.3. Analysis

Responses for each question were expressed as percentages of respondents for all categories. A Fisher’s exact test was used to compare the proportion of respondents reporting different training methods as most effective as a function of the type of training method they had themselves used on their dog. We adopted an alpha level of 0.05.

## 3. Results

As would be expected in a university-student population, most of the respondents were under the age of 30, with just 1.5% reporting that they were between the ages of 31 and 50. Women outnumbered men around three to one, which is typical of psychology majors at universities in the United States [23]. Seventy-one percent of respondents reported that the annual household income for the home where the dog resided was under $110,000. Average household size was 3.96 and, of those that reported the information, 39% of respondents reported that there were children under the age of 18 that resided in the home. Dog age ranged from around eight weeks old to over 17 years old. More male than female dogs were intact, but, for both sexes, most dogs were altered. Guardians acquired the dog from three common sources: rescue (27.6%), breeder (26.6%), or friends/family (23.5%).

In Spring 2021, when asked how the respondent would describe their relationship to their dog (Table 1), respondents primarily defined their relationship as either “friend” or “owner”. Fifteen respondents in Spring selected “other”, of which seven identified their relationship with the label “sibling”. Consequently, for the Fall 2021 survey, the option of “family member” was added and most students in that cohort selected this option and selection of “friend” and “owner” both declined.

Dogs were primarily walked on a flat collar and respondents reported that collars were commonly worn in the home as well (Table 2). Few respondents utilized aversive methods such as a chain or prong collar to walk their dogs.

Regarding what the dog wears inside the home, respondents could select more than one item to reflect dogs that might wear collars or harnesses in conjunction with a potential aversive device like an e-collar, invisible fence collar, or a citronella collar. Of the aversive collar options, e-collars to stop barking were the most reported but only by 2% of respondents.

When asked whether their dog was currently attending or had attended any training, in a group class, or with a private trainer, most respondents either reported that they did not attend any formal training or trained the dog themselves (Figure 1). Of those that did not receive any formal training, most respondents noted that they did not have the time, it was too expensive, or because their dog was perfect and did not, in their opinion, require it.

If they reported attending any type of formal training, 37.17% reported utilizing a trainer using mixed (aversive) methods and 60.20% sought a trainer who only utilized rewards-based (non-aversive) methods. For these dog guardians that identified their gender, 60% women guardians attended rewards-based (non-aversive) training compared to 59.7% of men; similar percentages were found for those who attended mixed (aversive) methods), 36.7% women and 35.7% men.

Of those who reported that their dog problematically pulled on leash, one third reported using some form of auditory correction to mitigate the problem followed by physical correction (Table 3). Of the 72% of respondents who reported leash pulling as a problem, very few used specialized leash equipment and all possible walking tools were utilized.

Across the remaining seven problem behaviors (Table 3), auditory and physical corrections were the most common techniques to mitigate the behavior. While e-collars worn inside the home for barking were only reported by 1.54% of respondents (Table 2), that rose to 2.85% when directly asked about mitigation of problematic barking.

Diversity of techniques was most apparent in response to aggression towards dogs and people. Interestingly, in managing aggression, respondents used physical and auditory correction for dog–dog aggression but incorporated time outs to mitigate aggression towards people. E-collars were reportedly used by 3.36% of respondents for dog aggression but were only used by 1.61% for human-directed aggression.

Beyond the relatively high percentage of reported use of time outs to manage aggression towards people, non-aversive methods such as managing the dog’s access to a triggering event or stimulus, ignoring the behavior, or rewarding alternative behaviors were not very popular. Ignoring the behavior was more popular in managing jumping and resource guarding, while rewarding an alternative behavior was the most reported technique to manage a dog that did not come when called.

When asked a general question about whom they would ask for advice, most respondents reported obtaining behavior information from online sources, followed by a veterinarian, and finally, friends or family (Table 4).

However, when asked about the eight specific problem behaviors, veterinarians were rarely reported as a recommending source, rather, friends and family were the most consistently reported information sources followed by online sources (Table 5).

When asked generally, 83.54% of respondents reported they did not use a book to obtain advice, but if they did, the most popular choices were books by Cesar Millan. Similarly, television was infrequently cited, but Cesar Millan’s The Dog Whisperer was the most popular show. The option of social media was added in the fall survey and, while infrequently reported, when asked generally and across the eight behaviors, if used, respondents either utilized TikTok or Instagram.

Only 5.03% of respondents across both semesters reported at least one use of an e-collar as a method to mitigate a problem behavior (Table 6). For those respondents, 62.5% reported that they either trained their dog themselves or attended no formal training. Of those who used an e-collar and attended formal training, “mixed methods” was reportedly the most effective technique, while “aversives-only” training was the least effective. Of those who used any form of physical correction for a problem behavior (not just e-collars), “mixed methods” were also reportedly the most effective approach. However, respondents who reported using auditory or physical corrections said that rewards-based methods were the most effective.

For both cohorts, all respondents reported that rewards-based (non-aversive) methods were most effective compared to the other training methods: either “mixed” or “aversives-only”. Examining who reported what method was most effective by gender, and percentages were quite similar; rewards-based (non-aversive) methods were most popular for women and men, 57.3% and 47.8%, respectively.

Of those who attended formal training (N = 192), the majority of respondents reported that whatever method they experienced in classes the most effective. However, respondents who trained with rewards-based methods reported that this method of training was more effective significantly more often than those who trained with mixed methods reported that mixed methods were most effective (Fisher’s Exact Test, *p* < 0.01, Figure 2).

When analyzing why respondents considered a specific training method most effective, “worked well with my dog’s learning style/motivation”, “better ability to control the dog”, and “dog does not show the behavior anymore” were the three most popular responses. This same trend was seen in respondents who attended formal training whether using mixed or non-aversive methods. For both, the most popular reason given was because the training method worked well for their dog’s learning style followed by the better ability to control the dog, and because the dog did not show the problem behavior anymore.

## 4. Discussion

The nearly 800 undergraduate student respondents in this study, given the choice, referred to their dog as a “family member”. This preference likely reflected best the relationship for these students since both the respondent and the dog may be of similar status as dependent entities within their household, but also indicated how survey structure could impact results. When “family member” was not an option, only seven students reported a relationship to that effect under “other” and most defaulted to potentially less representative choices such as “friend”, “owner”, or “parent”.

Problem behaviors, ranging from the mildly annoying like leash pulling to the potentially dangerous like aggression to other dogs or humans, were reported frequently: Only 2.8% of respondents reported that their dog did not exhibit any of the eight classes of problem behavior. Leash pulling might not require remediation, but aggression can be extremely dangerous and risky to handle without experience [21,22]. Notwithstanding how widespread problem behaviors were, most respondents did not utilize any professional training with their dog.

Much current concern in the literature focuses on the use of e-collars by pet owners. By not just asking general questions but also quite specifically about how respondents dealt with particular problems, we were able to uncover that aversive instruments were used more commonly than previously reported. For example, although only 1.5% of respondents reported having their dog wear a bark collar in the home, when asked specifically about problem barking, twice as many respondents (3%) reported using e-collars to mitigate that problem. While aversive collar use was still low, most respondents who utilized these tools reported that they either trained the dog themselves or did not seek out formal training.

Overall, when asked generally about mitigating their dog’s behavior, respondents provided what they may have considered were appropriate responses. For example, when asked broadly about whom they would reach out to for help with their dog’s problematic behavior, over 30% reported they would seek advice from their veterinarian. However, when asked specifically about the method they used to mitigate specific problem behaviors, veterinarians were reported as a source of guidance only 3.5% of the time—around one tenth as often. When asked about specific behavioral problems, respondents were much more likely to report obtaining advice from friends and family members or online sources. This decrease in using veterinarian-provided information may also relate to an observed trend where owners will obtain general information from their veterinarian, but further their understanding with research online [24].

The reported frequency of e-collar use found here is similar to rates reported by Blackwell et al. in the United Kingdom—ranging from about 1.7% in London to 7.3% in the eastern U.K. [17]. At the time of Blackwell and colleagues’ report, e-collars were banned in Wales and there were discussions on banning these devices in other regions of the United Kingdom. Masson et al. reported much higher rates, around 26%, in France [18]. As the authors noted, in addition to there being no comparable moves to ban e-collars in France, higher rates observed in their study could have been due to the wider audience they reached out to [18]. Blackwell and colleagues gathered respondents from dog-specific activities while the Masson study primarily gathered respondents from social networks [17]. Additionally, French owners noted that dogs not specifically acquired for companionship were more likely to wear an e-collar [18], whereas in our study, most dogs were considered family members, which might have reduced the frequency of e-collar use.

As seen in previous reports, non-aversive methods were the most popular approach among those who received professional training [17,19,21,25] and respondents who utilized non-aversive methods reported that they felt these were most effective compared to other methods [17]. Despite the majority of respondents here reporting use of non-aversive methods, aversive techniques were still regularly used. In this sample, auditory and physical corrections were the most common aversives reported, as also observed by Hiby et al. [21]. Of those in the current study who reported using rewards-based training (N = 115), only 11.3% did not use physical or auditory corrections to mitigate any of the eight problem behaviors; 21.7% reported using a physical or auditory correction to mitigate at least one of the behaviors and 6.1% reported using either physical or auditory corrections on their dog for all of the problem behaviors we asked about. This aligns with previous studies that note that aversive methods are used in combination with non-aversive methods [17,21,25,26,27].

There are several reasons why owners might recognize the benefit of non-aversive methods but fail to enact them in their daily lives. Todd noted that while the recent scientific literature indicates that the use of aversive methods may lead to poor welfare in pet dogs, these findings may not always be known or readily available to the public or even other animal professionals [28]. Although the American Veterinary Society of Animal Behavior has a position statement against the use of aversive methods [14], many accredited veterinary schools do not even offer a formal class on animal behavior [28], so it is hardly surprising that members of the general public may not be aware of the recommendation against aversive methods. A general veterinarian’s lack of behavior knowledge might also relate to why they were infrequently reported as a recommending source for any of the eight problem behaviors in this sample if guardians were informed or realized that their veterinarian did not have a comprehensive understanding of behavior.

Most respondents in our sample did not utilize professional sources for the methods they used to handle problem behaviors, a trend that is found in other studies and public surveys [20,22,25,29,30]. Of the 26% of French dog owners who reported using an e-collar, 75% purchased these devices either from the internet or a pet store and only 28.2% received a recommendation from a trainer or veterinarian [18]. As Hiby and colleagues noted, inexperienced individuals may have poor timing when delivering punishment, which can result in negative welfare outcomes [21]. Blackwell and colleagues found that there was an association between increased reported aggression and informal home training or no formal training, potentially highlighting the risk of training inexperience [20]. Thus, continuing to seek the advice of those without any education or experience puts dog owners, dogs, and others at risk [21].

In our sample, commonly reported reasons not to seek formal training were that respondents did not have the time or the money. Yet, if a problem exists, owners will still need a solution, one that may be more immediate or less expensive. People learn and develop their understanding and opinions from the people they socialize with and the information they are exposed to [21,28]. There is a wide range of sources of behavioral guidance available to dog owners which might be rife with misinformation. Browne and colleagues found numerous inconsistencies in best-selling dog training books on how they described basic learning theory principles and the promotion of training methods that might lead to poor welfare [31]. Even though television shows and books were infrequently used by guardians in this sample, when consulted, they were most frequently by Cesar Millan and viewed as a professional by the public notwithstanding the extensive critiques of his training methods [3,32]. The ready availability of the internet and social media may make it easier than ever for a dog guardian to obtain advice which may be inaccurate or potentially harmful to dogs [33]. When asked for their main source of pet health information, most pet owners reported that they searched the internet, but respondents recognized that the most trustworthy source was their veterinarian followed by friends and family members [34].

### Limitations

One limitation of this study was the nature of our study population. Young college students are not representative of the general U.S. population and were likely often not the primary caregivers for the dogs. They were, however, exposed to handling their dog in different situations in their daily interactions. Notwithstanding these limitations, our sample may be closer to the general dog owning population than those in studies that have recruited from populations more invested in the dog training world [19,20,21,35]. Additionally, despite recruiting a different sample population, our results fall in line with the previous literature [19,20,21,35], providing some support that our sample does provide insight into the general public.

Another limitation could be the possibility that respondents identifying as female preferred non-aversive methods and possibility skewed the sample since women trainers are more likely to utilize non-aversive methods over men [36]. However, percentages by gender for those who attended formal training were similar, as well as the percentages by gender who considered the respective training methods as most effective. This comparable use of aversive methods between female and male dog guardians falls in with previous research that found that female dog owners are in fact willing to use mild aversive methods [27].

Survey collection runs the inherent risk of self-report bias. While our undergraduate sample may not be as invested in the dog training world as the populations sampled in other studies, dogs and dog training often appear on across the internet and social media so it is not improbable that our respondents had an awareness of methods, particularly those utilizing physical punishment, that might appear “less acceptable” to researchers. Despite the possibility of report bias, both auditory and physical corrections were the most popular methods reported to mitigate problem behaviors reported here, so respondents either did not consider these as methods that could negatively impact dog welfare or considered them acceptable compared to more severe methods like e-collars. Similarly, our rates of reported punishment or reinforcement were in line with other survey studies [17,21,25,26,27].

Another limitation may have been in the description of training method types. We deliberately chose to describe them in broad, general terms so more specific training terminology did not confuse respondents. For example, we would label individuals who utilize both non-aversive to build behaviors and aversive methods to correct problematic ones only as using “mixed methods”, but a respondent might have viewed the infrequent use of aversives as “rewards-based”.

In our survey, we generally asked respondents about the presence or absence of a problem behavior; we did not seek information about how often this problem might occur as some previous studies have done, e.g., [21,26]. As noted by Rossi and colleagues, the existence of a problem behavior did not always relate to consistent use of a certain method to mitigate that behavior [26]. As a result, when it does happen, are owners always consistent in executing a method to handle the problem or does that method change based on context? Additionally, structured like other similar studies, e.g., [20,21,22], we acknowledge that guardians would seek the advice of multiple individuals as well as trying different methods to mitigate problem behavior. A more optimal survey design might be to ask guardians to rank mitigation methods, how often they would use them and if they received the recommendation for multiple sources.

## 5. Conclusions

This study sampled university undergraduate students to derive an understanding of the methods and training tools that dog guardians in the United States use. Very few respondents sought the assistance of a professional dog trainer when they had specific concerns about their dog’s behavior and often sought advice from friends, family, or online sources. Most respondents, when asked generally, claimed that they used rewards-based methods, but in relation to specific problem behaviors, most reported using some form of aversive method. Given the severity of some behavior problems like aggression and the prevalence of inaccurate and misleading information about dog training, not seeking advice from trained professionals presents a risk to pet dog welfare and the families that share their homes.

## Figures and Tables

**Figure 1 animals-14-01310-f001:**
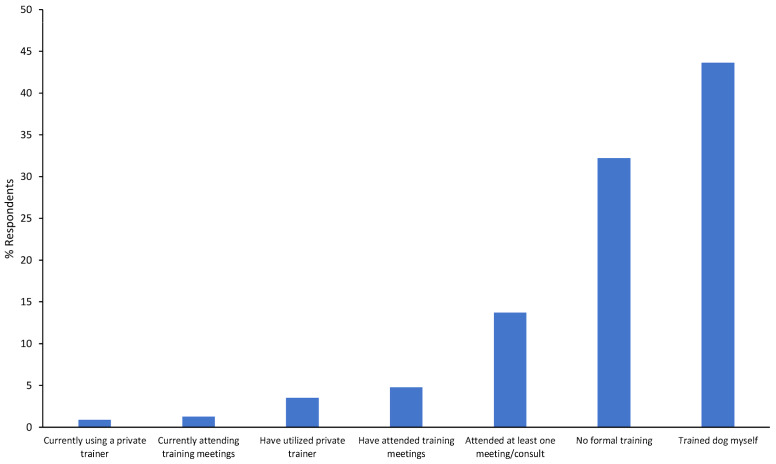
Survey responses of participants from both semesters of training experience, N = 795.

**Figure 2 animals-14-01310-f002:**
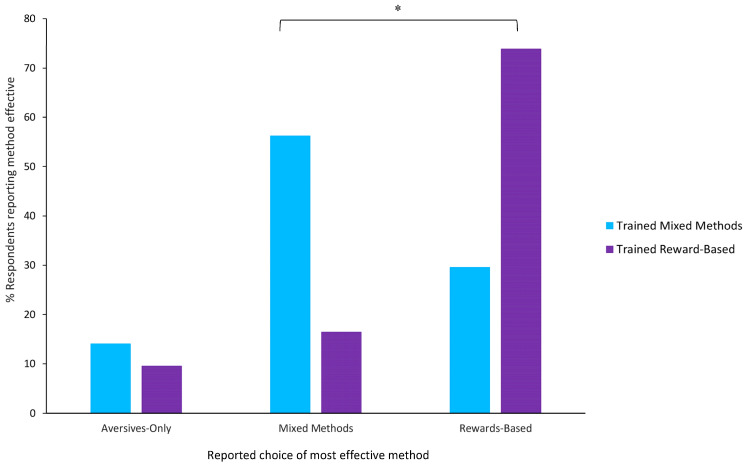
Most reported effective method, summary across both semesters, of respondents using different methods: * *p* < 0.001. Note. Data for those who experienced aversives-only training is not shown since only four participants reported experiencing that form of training.

**Table 1 animals-14-01310-t001:** Survey responses pertaining to how participants describe their relationship to the dog, Spring 2021 N = 398, Fall 2021, N = 398.

Relationship	Spring 2021 Percentage	Fall 2021 Percentage
Family Member	----	47.5
Friend	38.7	17.1
Guardian	8.0	5.5
Master/Mistress	0.5	0.3
Owner	27.9	17.3
Parent	21.1	11.6
Other	3.8	0.8

**Table 2 animals-14-01310-t002:** Survey questions and responses pertaining to training tool use.

What do you walk your dog on? N = 796		
	Equipment	Percentage
	Back clip harness	37.19
	Chain collar	2.76
	Flat collar	43.59
	Front clip harness	2.51
	Head halter	2.64
	Martingale	3.64
	Prong collar	0.26
	Not walk dog	5.03
What does your dog wear inside the home? Select all that apply. N = 842		
	Activity monitor	0.24
	Citronella collar	0.24
	Collar	54.87
	E collar barking	1.54
	E collar invisible fence	0.05
	Harness	1.54
	No collar (naked)	41.09

**Table 3 animals-14-01310-t003:** Percentage of survey responses pertaining to the problem behavior and the respondents’ chosen method to mitigate the problem. Grand total of respondents = 796.

	Auditory Correction	Ignore	Manage Dog’s Access	Physical Correction	Reward Alternative Behavior	Time Outs	Use of e-collar	Use of Specialized Leash Equipment *	Not a Problem My Dog Has	No Response
Barking	58.79	10.46	12.36	3.80	6.97	4.75	2.85	-	19.72	1.00
					total with problem			631		
Destructive Behavior	37.75	1.25	11.50	12.25	7.00	29.75	0.05	-	48.74	1.00
					total with problem			400		
Dog Aggression	33.81	2.88	18.22	26.37	5.99	9.35	3.36	-	46.48	1.13
					total with problem			417		
Jumping	41.98	15.64	3.24	22.90	9.73	5.53	0.95	-	33.16	0.87
					total with problem			524		
Leash Pulling	32.79	10.57	10.93	21.86	9.85	-	1.25	12.72	28.77	1.13
					total with problem			558		
Not Come when Called	44.47	9.33	1.97	5.15	31.94	5.65	1.47	-	47.73	1.13
					total with problem			407		
People Aggression	33.54	1.29	9.67	32.26	6.45	26.12	1.61	-	59.92	1.13
					total with problem			310		
Resource Guarding	39.88	11.78	10.88	15.40	14.50	6.94	0.60	-	57.16	1.25
					total with problem			331		
* The nature of the specialized equipment used by respondents who selected that pulling on a leash was a problem and also selected that they mitigated the problem with “specialized leash equipment” N = 70										
	Back Clip Harness	Chain Collar	Flat Collar	Front Clip Harness	Head Halter	Prong Collar				
	48.57	11.43	12.86	2.86	7.14	17.14				

**Table 4 animals-14-01310-t004:** Survey questions and answers pertaining to where respondents received their training advice.

If you have a problem with your dog, who do you ask for advice? N = 796		
	Source	Percentage
	Breeder or shelter where I got the dog from	1.38
	Dog Trainer	4.77
	Friend/Family	23.87
	Online	36.31
	Social Media ^a^	2.76
	Veterinarian	31.53
	No response	0.75
What book do you use for advice? N = 796		
	Book	Percentage
	Brandon McMillian	1.51
	Cesar Millan	7.29
	Monks of New Skete	2.39
	Sophia Yin	1.38
	Zac George	1.38
	Other	1.88
	None	83.54
	No Response	0.63
What television show do you use for advice? N = 796		
	Television	Percentage
	Canine Intervention ^a^	5.58
	Dog Impossible	1.13
	It’s Me or the Dog	3.02
	Lucky Dog	6.66
	The Dog Whisperer	23.38
	Other	5.77
	None	52.51
	No Response	4.77
What social media sites/apps do you use for behavior advice? N = 398		
	Social media	Percentage
	Facebook	17.59
	Instagram	30.90
	TikTok	38.19
	Twitter	5.53
	No Response	7.79

Note: ^a^—The social media question was added for the Fall 2021 survey, percentage presented reflects only Fall responses.

**Table 5 animals-14-01310-t005:** Percentages of who respondents ask for the implemented mitigation method for problem behavior.

	Book	Friends/Family	Online	Social Media ^a^	Trainer	Television	Veterinarian	Other
Barking N = 614	0.98	51.14	14.82	3.86	14.17	3.75	3.75	11.40
Destructive behavior N = 398	1.01	45.48	16.58	4.04	16.83	3.27	5.78	9.05
Dog aggression N = 412	0.73	44.42	15.29	2.87	17.23	5.10	6.07	9.71
Jumping N = 520	2.31	45.96	15.58	5.77	16.35	4.04	2.88	10.00
Leash pulling N = 553	0.36	43.58	16.82	4.49	17.36	4.88	4.34	10.48
Not come when called ^b^ N = 349	2.87	45.27	13.75	3.40	20.63	4.58	3.44	7.45
People aggression N = 398	1.62	45.48	16.58	4.04	16.83	3.27	5.78	9.05
Resource guarding ^b^ N = 280	1.43	48.21	16.43	4.55	15.71	4.29	5.36	6.07

Note: ^a^—The social media question was added for the Fall 2021 survey, percentage presented reflects only Fall responses. ^b^—For some respondents in Spring 2021, skip logic was not working so for about 100 respondents (estimated 50 per problem) with dogs for these problems did not receive a prompt to who gave the recommendation.

**Table 6 animals-14-01310-t006:** Survey response analysis of those using aversive correction methods.

Summary for unique responses of at least one use of e-collar across any of the eight problem behaviors N = 40		
	Training experience	Percentage
	Trained self or no formal training	62.50
	Mixed methods	25.00
	Rewards-based	12.50
Respondents who report use of e-collar—what training method most effective N = 40		
	Training method	Percentage
	Mixed methods	55.00
	Rewards-based	17.50
	Aversives-only	27.50
Respondents who report use of physical correction across any of the eight problem behaviors—what training method most effective N = 596		
	Training method	Percentage
	Mixed methods	42.11
	Rewards-based	40.10
	Aversives-only	17.79
Respondents who report use of physical or verbal correction across any of the eight problem behaviors—what training method most effective N = 2079		
	Training method	Percentage
	Mixed methods	39.73
	Rewards-based	46.75
	Aversives-only	13.41

## Data Availability

The data presented in this study are available on request from the corresponding author.

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
