# Peer review of "Training Methods Used by Dog Guardians in the United States: Prevalence, Sources of Information, and Reasons for Use"

_animals, 2024, doi:10.3390/ani14091310_

Round 1
Reviewer 1 Report (Previous Reviewer 1)
Comments and Suggestions for Authors
The authors have adequately addressed my concerns. Thank you.
Author Response
Thank you for accepting the revisions.
Reviewer 2 Report (New Reviewer)
Comments and Suggestions for Authors
The topic of use of training methods is an important one in the context of better understanding of human-canine interactions and dog welfare in general. The background introduction provides a good overview of some of the issues. However, this study has serious limitations. The authors themselves note that their sample of “young college students” is not likely to be representative of the US population or dog owners in general, but minimize this flaw. The sampling issue is not limited to age alone. Their study population is also significantly skewed in gender (75% female), as well as likely unrepresentative in income and possibly ethnicity. They give no data examining gender as a variable in their findings, despite the fact that women might be reasonably hypothesized to have very different opinions and actions on the use of aversive vs. non-aversive training methods, a major focus of the study. This is a significant oversight.
In listing potential sources of information, they omit animal shelter/humane society staff as a significant potential source of data despite the growing number of shelters that employ behavior staff that often have certification (e.g. APDT) or other professional training. The study asked questions about the source of the dog being reported on, but this result is not reported. This also could impact attitudes towards training and the selection of sources of information.
The discussion section missed opportunities for evaluation of several key issues. They note that 31% of respondents would ask a veterinarian for advice on a problem, although generally fewer than 6% actually asked vets for advice on specific behavior problems. This could reflect the fact that, despite significant growth in interest in veterinary behavior issues, vets in general still receive little training in specifically addressing behavior issues. There is a disconnect between the “halo effect” of assuming that vets are all-knowing in dog-related issues and the reality of the limited experience of most vets other than those with specific veterinary behavior training.
The reliance of a significant proportion of the respondents on social media as a source of information on addressing training deserves more comment and discussion, particularly with concern about the quality of these sources as providing notoriously bad advice on other issues such as nutrition and health in general. Also, this high level of reliance on social media is likely to be an artifact of the largely young, female demographic of the sample.
The relatively high reliance of the works of Cesar Millan as a potential source of training information also deserves further comment in light of the widespread criticism of Millan from veterinary, dog training and animal welfare professionals. See for example:
https://www.nytimes.com/2006/08/31/opinion/31derr.html “Pack of Lies”
Mech, L.D. 2008. What ever happened to the term ‘alpha wolf’. International Wolf, 4-8. www.wolf.org
Luescher, Andrew. “Letter to National Geographic Concerning ‘The Dog Whisperer.’” Weblog Entry. Urban Dawgs. Accessed on Novermber 6, 2010. (http://www.urbandawgs.com/luescher_millan.html)
Kerkhove, Wendy van. “A Fresh Look at the Wolf-Pack Theory of Companion Animal Dog Social Behavior” Journal of Applied Animal Welfare Science; 2004, Vol. 7 Issue 4, p279-285, 7p.
Jackson-Schebetta, L. (2009). Mythologies and commodifications of dominion in The Dog Whisperer with Cesar Millan. Journal for Critical Animal Studies, 7(1), 107-31
Author Response
All revision comments are addressed in the attached document.

Round 2
Reviewer 2 Report (New Reviewer)
Comments and Suggestions for Authors
Most of the suggested changes have been incorporated into the revised manuscript.
Although the research questions are interesting and the survey is appropriately designed, the basic shortcoming of a highly selective sample (college psychology students) that is non-representative of the general population of dog owners cannot be overcome without surveying an entirely new population.
It is still of value in attempting to illuminate questions related to how dog training decisions are approached.
This manuscript is a resubmission of an earlier submission. The following is a list of the peer review reports and author responses from that submission.
Round 1
Reviewer 1 Report
Comments and Suggestions for Authors
Johnson and Wynne’s Training methods used by dog guardians in the United States: Prevalence, sources of information, and reasons for use, manuscript ID animals-2919212
Johnson and Wynne address a very important issue of how dog guardians punish or reward behaviors that the guardians find inappropriate. Given their results on how common the use of punishments is, this manuscript might make an important contribution to the literature and hopefully to the welfare of many dogs. Unfortunately, the manuscript is missing its introduction (only the paragraph in the template describing what should be included in an introduction is present), it is difficult to determine whether the manuscript is contributing new results for this area of research. Also, the sample is not representative of the population and likely was not living with their dog at the time the questionnaire was answered. This reduces the ability to generalize the results to people who are living with and changing the behaviors of a dog.
Issues:
The introduction is missing from the manuscript that I downloaded. Without it, it is difficult to decide if the manuscript is making a new contribution to the literature. Given the references, I believe that an introduction exists. Why it was not included in the submitted manuscript is a different question.
This would be a much stronger paper if the data had been collected from a sample of people who were currently living with a dog and possibly trying to change the dog’s behavior. The limits of the sample are partially addressed in the limitations section of the discussion, but I believe that the sample used greatly reduces the ability to generalize the results and perhaps brings into question the validity of the results.
I think that perhaps “training” is not the appropriate word to use throughout the manuscript. “Training”, in my opinion, refers to the process of learning the basic commands that dogs should know such as sit, stay, come, etc. I’m probably wrong about that. Most of the manuscript seems to be more focused on what type of punishment (which never should be used) or reward is used in an attempt to change behavior. “Behavioral modification” might be a better term than “training”.
Table 3: Given that the percentage of people using a particular method for a particular behavior sums to close to 100%, I assume that respondents could select only one method of correction. If that assumption is true, does allowing people to select a single method of correction reflect how people correct their dogs? For example, couldn’t someone use both an auditory correction and then follow up with a reward for a more desirable/redirected alternative behavior?
Table 3: Table 3 is needlessly long. Much of it could be reorganized into a matrix such as is suggested for table 4 below.
Table 4: Like table 3, I assume that each participant could select a single answer for each question. Does this reflect what people do? Couldn’t a person ask a family member or friend, look online, consult social media, and consult their veterinarian?
Table 4: Some of the percentage for a given question sum to over 100% (even with roundoff considered). For the question about where the person gets information, the percentages sum to 101.37%. Other questions in table 4 also have percentages that sum to more than 100%.
Table 4 is needlessly long. Much of it could be reorganized:
|
|
Book |
Friend / Family |
Online |
Social Media |
Trainer |
TV |
Vet |
Other |
|
Barking |
|
|
|
|
|
|
|
|
|
Destructive behavior |
|
|
|
|
|
|
|
|
|
Dog aggression |
|
|
|
|
|
|
|
|
|
Jumping |
|
|
|
|
|
|
|
|
Lines 189-190: This sentence does not make sense. Should “of” be “have”?
In the supplemental material, question 24 does not include “none” as an option, but table 4 indicates that 52.51% of the participants selected “none” as the response to this question.
Author Response
Please find the attached document that highlights (in green) our responses to the concerns previously noted. The updated document has been re-submitted with the appropriate introduction

Reviewer 2 Report
Comments and Suggestions for Authors
Regrettably, it appears that the version of the manuscript uploaded is not the final iteration. Not only is the introduction missing, but also the comments by the senior researcher/supervisor remain in its place. Due to this discrepancy, I refrained from reading the remainder of the manuscript as it appears to be incomplete.
Comments on the Quality of English LanguageFrom what I've read so far, the quality of English language appears to be quite high.
Author Response
We have provided a resubmitted document with the introduction included. We apologize for our previous oversight.
Reviewer 3 Report
Comments and Suggestions for Authors
Dear authors,
thanks for the paper, which supports the notion that owner education in topics as dog behaviour, learning theory and training, and welfare is still very much needed.
Author Response

(The authors gave the same response as above.)
